# A deep learning method for replicate-based analysis of chromosome conformation contacts using Siamese neural networks

Ediem Al-jibury [1,2] ✉, James W. D. King[1], Ya Guo[1,3,4], Boris Lenhard [1,5], Amanda G. Fisher [1], Matthias Merkenschlager [1,7] ✉ & Daniel Rueckert [2,6,7] ✉

The organisation of the genome in nuclear space is an important frontier of biology. Chromosome conformation capture methods such as Hi-C and Micro-C produce genome-wide chromatin contact maps that provide rich data containing quantitative and qualitative information about genome architecture. Most conventional approaches to genome-wide chromosome conformation capture data are limited to the analysis of pre-defined features, and may therefore miss important biological information. One constraint is that biologically important features can be masked by high levels of technical noise in the data. Here we introduce a replicate-based method for deep learning from chromatin conformation contact maps. Using a Siamese network configuration our approach learns to distinguish technical noise from biological variation and outperforms image similarity metrics across a range of biological systems. The features extracted from Hi-C maps after perturbation of cohesin and CTCF reflect the distinct biological functions of cohesin and CTCF in the formation of domains and boundaries, respectively. The learnt distance metrics are biologically meaningful, as they mirror the density of cohesin and CTCF binding. These properties make our method a powerful tool for the exploration of chromosome conformation capture data, such as Hi-C capture Hi-C, and Micro-C.

Eukaryotic chromatin is spatially organised within the cell nucleus to facilitate essential genome functions including transcription, replication, repair, and chromosome segregation[1–6]. Mutations that affect nuclear architecture can lead to disease[7–9]. Aspects of this organisation such as the formation of chromosome territories, the separation of heterochromatin from euchromatin, and the formation of topologically associating domains (TADs) are highly conserved[10], but finer details such as chromatin loops and the strength of contact domains

can vary in a cell type-specific manner[11]. While the precise mechanisms governing chromatin organisation remain an intense area of study, one of the main mechanisms for the formation of domains and loops is the active extrusion of chromatin by the cohesin complex, which is constrained by CTCF binding[12–15].

Key tools for mapping the organisation of genomes include Hi-C and Micro-C[16,17]. Both combine proximity-based ligation and high throughput sequencing to produce genome-wide chromatin contact

[1]MRC LMS, Imperial College London, London W12 0NN, UK. [2]Department of Computing, Imperial College London, London SW7 2RH, UK. [3]Sheng Yushou Center of Cell Biology and Immunology, Joint International Research Laboratory of Metabolic and Developmental Sciences, School of Life Sciences and Biotechnology, Shanghai Jiao Tong University, Shanghai 200240, China. [4]WLA Laboratories, Shanghai 201203, China. [5]Sars International Centre for Marine Molecular Biology, University of Bergen, 5008 Bergen, Norway. [6]Klinikum rechts der Isar, Technical University of Munich, 81675 Munich, Germany. [7]These authors jointly supervised this work: Matthias Merkenschlager, Daniel Rueckert. ✉e-mail: e.aljibury@lms.mrc.ac.uk; matthias.merkenschlager@lms.mrc.ac.uk; d.rueckert@imperial.ac.uk

maps that provide a rich data source containing quantitative and qualitative information about genome architecture. The analysis of Hi-C and Micro-C data typically relies on visual inspection combined with methods that score known features of chromatin conformation maps, for example insulation score, directionality index, TADs, loops or stripes[1,10,18]. This approach is not only laborious but could also result in features of biological importance being missed. New methods are required for the analysis of the vast quantities of information contained in genome-wide chromatin conformation capture data.

One approach, well suited to the large levels of data, is to use deep learning to find differences in chromatin conformation data[19]. Here, the problem becomes identifying subtle differences between a largely heterogeneous genome and accounting for very high levels of non-uniform noise present in conformation contact maps. Viewing these conformation maps as images allows the use of image analysis techniques. Hi-C and Micro-C data contain high levels of noise that are not present in standard image datasets, and the suitability of naive image similarity metrics for the analysis of such data is currently under discussion[20,21]. To address this issue we have developed and validated

Twins, a deep learning-based analysis method which leverages replicates using a Siamese convolutional neural network[22]. Unlike standard image analysis tools, Twins learns to distinguish technical noise (differences between replicates) from biological variation (differences between conditions) using contrastive learning[23]. We validate the resulting embedding distance metric as biologically meaningful using independent chromatin immunoprecipitation data, and demonstrate the robustness of our method to technical noise associated with Hi-C normalisation and sequencing depth. Finally, we use the trained Twins network to extract features which are gained or lost between conditions in Hi-C maps.

## Results

### Twins learns to distinguish technical from biological variation

We developed a Siamese network metric learning approach from chromatin conformation data that takes advantage of Hi-C replicates in order to distinguish technical noise from chromatin conformation changes between biological conditions (Fig. 1a, b). Identical convolutional neural networks with shared weights are trained to produce an

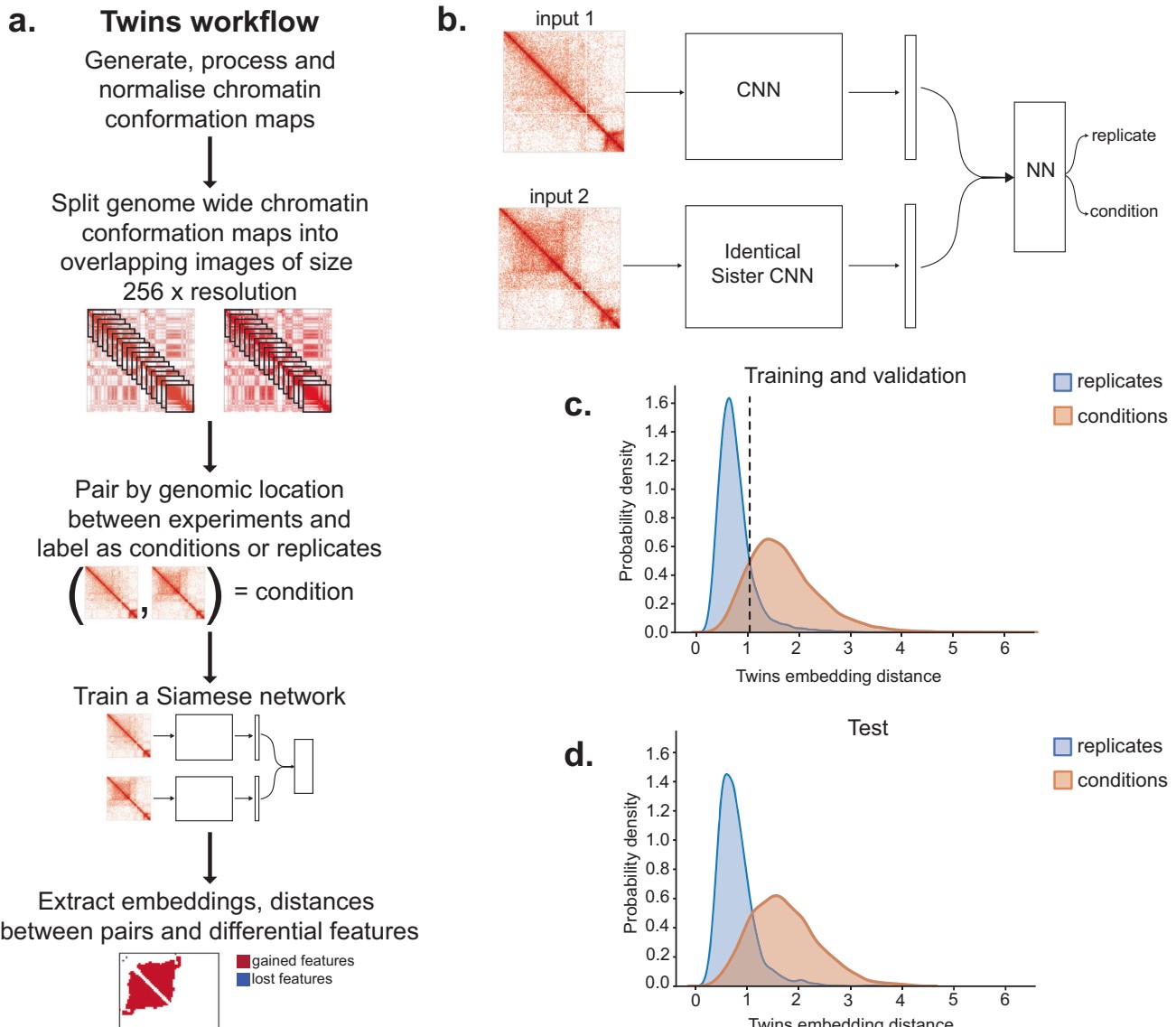

**Fig. 1 | A replicate-based method for deep learning from chromatin conformation contact maps. a** Overview of the Twins workflow. **b** Illustration of the Siamese network architecture for chromatin conformation maps. **c** Twins embedding distance distributions for replicates (blue) and conditions (orange, mature versus immature thymocytes) for training (chromosomes 1, 3–17, 19) and validation (chromosome 18). **d** As in (**c**) but on test set (chromosome 2).

embedding for regions taken from Hi-C paired by genomic location. Initially, we applied this approach to Hi-C data generated from immature (CD4 CD8 double positive, DP) and mature (CD4 single positive, SP) T lymphocytes derived from mouse thymus on each of two replicates per cell type. We split each Hi-C map into overlapping regions along the diagonal with a stride of 160 kb, measuring 2.56 Mb in size at 10 kb resolution normalised by Knight-Ruiz (KR)[24]. We paired the resulting images across the genome. Each pair was labelled either as a replicate pair (i.e. a pair of images taken from two biological replicates under the same biological condition) or a condition pair (i.e. a pair of images taken from two different biological conditions). The network was then trained using contrastive learning to minimise the Euclidean distance between replicate pair representations and maximise the distance for condition pair representations. An additional cross embedding loss term was added at the end of on fully connected layer as a regularisation term. We omitted chromosomes 2 and 18 from training in order to maintain independent test and validation sets. We used performance on chromosome 18 to avoid over-fitting by halting training when the validation loss began to increase. We find our network learns to separate the variation from the replicate noise (Fig. 1c). To assess our network performance we used two metrics which judge the separability of the replicate and condition pair distance distributions; mean performance and the separation index each vary from 0 (overlapping) to 1 (well separated). We apply these measures on the remaining and previously unseen test chromosome 2 and find that embedding distances for different biological conditions are well separated from replicates with a separation index of 0.7001 and a mean performance of 0.8513 (Fig. 1d). To demonstrate that these results are independent of the choice of test chromosome, we trained new networks each with a different choice of test chromosome omitted from training selected from 1 to 6. We found that the resulting embedding distances were highly correlated genome wide (Fig. S1).

## The Twins training procedure leads to a relevant distance metric

We then trained our network independently on two additional datasets; hepatocytes under control conditions (tamoxifen control) versus deletion of the *Nipbl* cohesin gene (ΔNIPBL)[3] and neural progenitor cells under control conditions compared to CTCF degradation (CTCF degron)[25]. In each case the separation index and mean performance remained high (Fig. 2a). Figure 2 shows the resulting embedding distances for chromosome 2. Deletion of Nipbl resulted in extensive separation between biological conditions and replicates across most of chromosome 2 (Fig. 2b). This was expected, as NIPBL is required for cohesin loading, and the loss of cohesin from chromosomes abrogates the formation of contact domains and loops[3,15,26]. CTCF degradation also led to extensive changes, but compared to ΔNIPBL these changes were less uniformly distributed along the length of the chromosome (Fig. 2c). This is consistent with the reported focal loss of insulation at domain boundaries in response to CTCF degradation, but the preservation of domains[2]. Compared to the global loss of NIPBL/cohesin and CTCF, T cell differentiation was associated with regionally selective changes in embedding distances, which likely represent differentiation-associated changes in 3D genome organisation (Fig. 2d).

To challenge the ability of the network to recognise an erroneous input, we applied the trained Twins ΔNIPBL metric to a shuffled data set contained an equal number of reads derived from ΔNIPBL and the corresponding control Hi-C data[20]. The scores for the shuffled data combinations were similar to replicate pairs, and clearly different from condition pairs (Fig. S4). The ability of the network to dismiss the shuffled data demonstrates the robustness of the Twins metric.

To test the validity of the Twins metric distributions generated for ΔNIPBL and CTCF degron, we compared the Twins metric to cohesin and CTCF binding sites as determined by chromatin immunoprecipitation. To quantify whether Twins distances reflect the biological

perturbations anlysed, we separated the genome into regions with high versus low density of cohesin and CTCF binding sites as determined by chromatin immunoprecipitation followed by high throughput sequencing (ChIP-seq) (Fig. S2). Consistent with the essential role of NIPBL in the chromosomal loading of cohesin[27], genomic regions with a high density of ChIP-seq peaks for the cohesin subunits RAD21 and SMC3 showed significantly higher Twins distances in ΔNIPBL than genomic regions with a low density of RAD21 and SMC3 peaks (Fig. 2e, $p = 1.4e-191$ and $p = 2.9e-230$)). Genomic regions with a high density of CTCF ChIP-seq peaks showed significantly higher Twins distances in CTCF degron than genomic regions with a low density of CTCF peaks (Fig. 2f, $p = 2.3e-37$) whereas regions with a high and low number of H3K27me3 peaks did not show any significant change in distance distribution 2f, $p > 0.05$) consistent with previous studies[2]. Taken together, these data demonstrate the validity of the Twins metric.

In the T-cell differentiation system, we observed that regions with visual changes in chromatin conformation had high embedding distances, while regions without visible changes in chromatin conformation had low embedding distances (Fig. S3a). Quantifiable changes in Hi-C features such as the gain or loss of contact domains, altered A/B compartmentalisation, directionality, and insulation resulted in significantly higher embedding distances (Fig. S3b). To test the sensitivity of Twins we applied our trained T-cell differentiation network to Hi-C data from DN2 thymocytes where deletion of the distal Bcl11b enhancer results in localised chromosome conformation change[28]. Twins scores across chromosome 12 showed a prominent peak centred around the location of the deleted Bcl11b enhancer at 108.4 Mb (Fig. 2g). This demonstrates that the Twins score provides a good reflection of visual changes to the Hi-C map and can indeed identify small-scale differences that result from focal perturbation of the genome.

Finally, we retrained the hepatocyte ΔNIPBL network to include additional ΔNCAPH2 Hi-C[29]. The inclusion of the ΔNCAPH2 did not affect the overall distributions of the ΔNIPBL data (Fig. 3a). However, Twins found no clear differences in embedding distances between ΔNCAPH2 and control hepatocytes either visually along chromosome 2 (Fig. 3b) or quantitatively (Fig. 3c). This was reassuring for two reasons. Firstly, changes in chromosome conformation in response to Ncaph2 deletion are known to become visible only after the completion of at least one cell cycle, and hepatocytes are largely quiescent in vivo[29]. Secondly, this demonstrates that the use of replicates prevents the network from learning arbitrary differences even with enforced contrastive loss learning.

## Comparison of Twins with naive image similarity metrics

To assess the performance of the Twins metric compared with a naive metric, we applied five well-established image similarity metrics to Hi-C data generated from immature DP and mature CD4 SP thymocytes on each of two replicates per cell type. Two of these metrics, SSIM and PSNR, have recently been used in the analysis of Hi-C data[20,21,30]. As described above, KR-normalised Hi-C maps at 10 kb resolution were split into regions measuring 2.56 Mb in size, and the resulting images were paired by genomic location.

In contrast to Twins, across chromosome 2, replicates and conditions are not well resolved by the five image similarity metrics (Fig. 4a). This was the case even for regions that undergo visible reorganisation during the course of thymocyte differentiation. To illustrate this, we selected two example regions on chromosome 2. The first is a noisy replicate pair without actual differences in chromatin conformation. The second region contains a contact domain that is lost during thymocyte differentiation (chr2: 48.5–51 Mb). Here, the loss of a contact domain was associated with the developmentally regulated change in the expression of the *Mmadhc* gene. The SSIM and PSNR metrics failed to understand inherent noise in the Hi-C map and misclassified this noise as high levels of image dissimilarity (Fig. 4b),

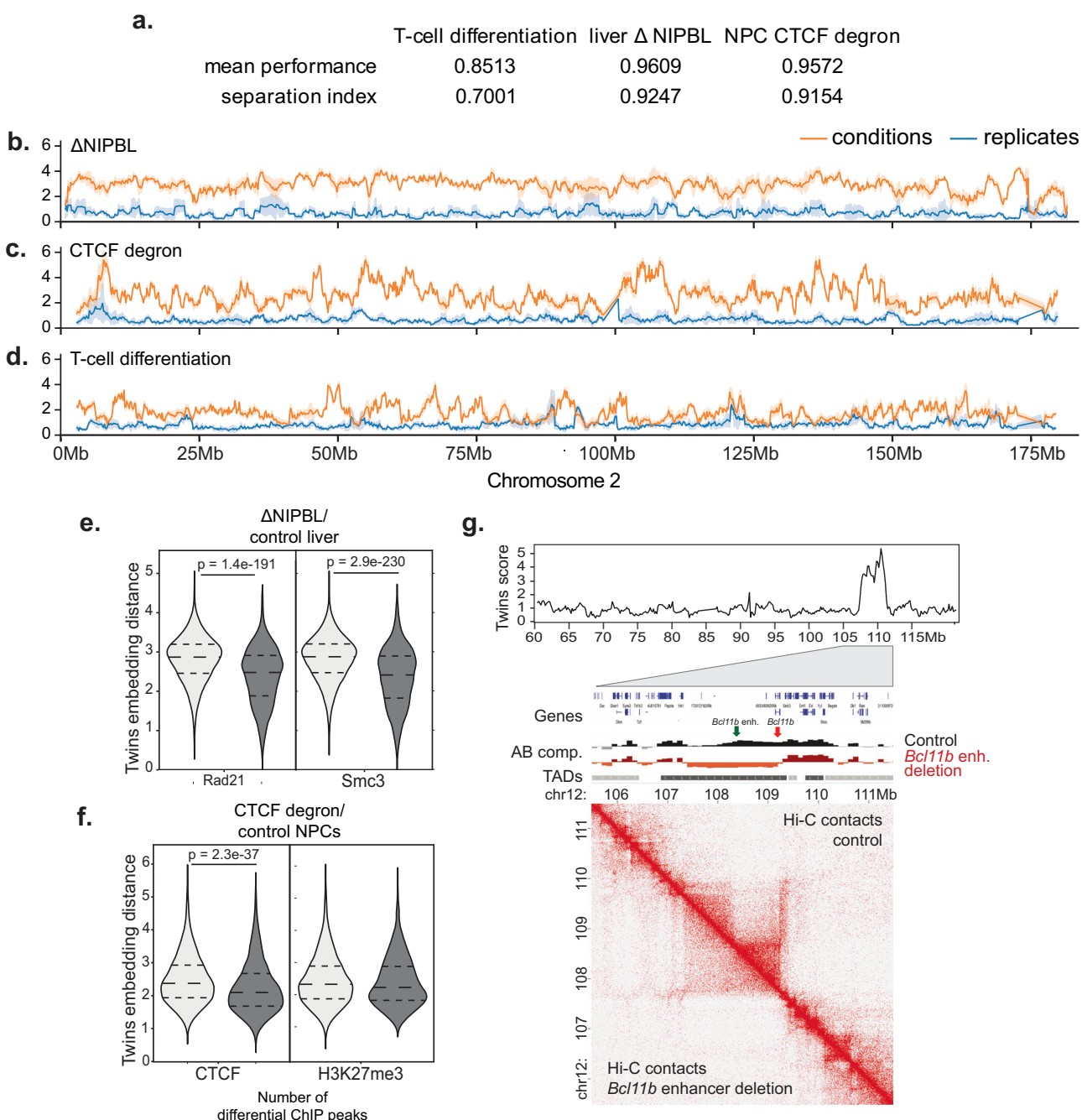

**Fig. 2 | Twins delivers meaningful embedding distances for a range of different biological settings. a** Mean performance and separation index for independent Twins networks on three datasets; T-cell differentiation, ΔNIPBL and CTCF degron. **b** Normalised Hi-C maps at 10 kb resolution generated from ΔNIPBL versus control hepatocytes were split into regions measuring 2.56 Mb in size, and the resulting images were paired by genomic location. Twins embedding distances for conditions (orange) and replicates (blue) are plotted across test chromosome 2. Data are presented as mean values ± the 95% confidence interval. **c** As in (**b**) for neural progenitor cells under CTCF degron versus control conditions. **d** As in (**b**) for thymocytes at CD4 SP versus DP stages of differentiation. **e** Regions are categorised by the density of ChIP-seq peaks. For ΔNipbl data, regions are split into high and low ($n = 2063, 13,154$, respectively) and Twins scores are calculated for the two sets.

Regions with high levels of RAD21 binding have higher Twins scores by two-sided t-test ($p = 1.4e − 191$). For Smc3 binding, regions are split into high and low ($n = 1813, 13,404$, respectively). Regions with high levels of Smc3 binding have higher Twins scores by two-sided t-test ($p = 2.9e − 230$). **f** For the CTCF degron data, regions are split into those with high and low CTCF binding ($n = 4192, 10,043$, respectively) and those with high and low H3K27me3 binding ($n = 1312, 12,923$, respectively). Regions with high levels of CTCF binding had significantly higher Twins scores by t-test ($p = 2.3e − 37$). but regions with high levels of H3K27me3 binding did not have significantly different Twins scores by t-test ($p = 0.053$). **g** The results of the T-cell network applied to an enhancer deletion in DN2 thymocytes. The Twins network is able to identify the region containing the enhancer deletion as having a differential chromatin conformation.

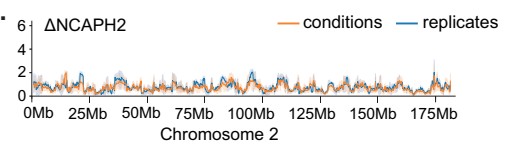

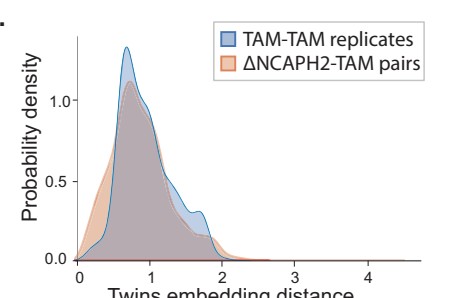

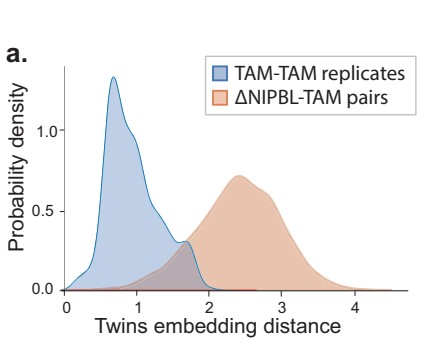

**Fig. 3 | Twins network does not find differences after NCAPH2 ko. a** Twins embedding distance distributions for replicates (blue) and ΔNIPBL versus control hepatocytes (orange) on network trained to include ΔNCAPH2. **b** For ΔNCAPH2 versus control hepatocytes, Twins embedding distances for conditions (orange) and replicates (blue) are plotted across test chromosome 2 using a network trained with ΔNCAPH2 in addition to ΔNIPBL and tamoxifen control data. Data are presented as mean values ± the 95% confidence interval. **c** As in (**a**) but with ΔNCAPH2 versus control hepatocytes (orange).

while the MSE, Hausdorff and NMI metrics were unable to score differences between Hi-C regions of DP and CD4 SP thymocytes (Fig. 4b). We calculated the separation index and mean performance between the distance distributions for replicate pairs and condition pairs for each the five standard image similarity metrics and the learnt Twins metric in thymocyte differentiation, CTCF degron, and ΔNIPBL. The learnt Twins metrics was far more capable of distinguishing technical noise from biological differences than the five established image similarity metrics in each of the systems (Fig. 4c).

### Twins is robust to Hi-C normalisation methods

To address the impact of Hi-C normalisation approaches we applied four of the most widely used normalisation method: Knights-Ruiz (KR)[24], Vanilla coverage (VC), square root of vanilla coverage (VC SQRT)[1] and iterative correction (ICE)[31] to Hi-C data for thymocytes at CD4 SP versus DP stages of differentiation. For each normalisation we trained a Twins network and then produced distances for each genomic location along the test chromosome. Across chromosome 2, the distance distribution was visually highly similar regardless of the normalisation (Fig. 5a). We also found a high correlation coefficient for all normalisations used ($p < 2e-308$, Fig. 5b), indicating that the method applied for the normalisation of Hi-C data does not impact Twins performance.

### Twins has a data set-dependent optimal operating resolution

Next, we looked at the effect of the chosen Hi-C resolution on the separability of embedding distributions for conditions and replicates. In this case, for each resolution $R = 2$, 5, 10 and 25 kb the genome was split into overlapping windows of size 256$R$. For mouse thymocyte differentiation data at an average sequencing depth of 234 M contacts, we found that the condition and replicate distributions were optimally separated at a resolution of 10–25 kb (Fig. 5c). We hypothesised that this is a product of the sequencing depth of the data. To test this, we applied Twins to a Micro-C dataset comparing human H1 and HFF cell-lines in human[32] at 365.46 M contacts (Fig. S5). Twins performed noticeably better on Micro-C than Hi-C data. There are several possible reasons for this. Micro-C has higher resolution, and therefore contains many more fine-scaled contacts. The increased resolution of Micro-C also enables near-optimal performance of the Twins network across a range of window sizes and resolutions (Fig. 5d). In addition, human embryonic stem cells (H1)

and fibroblasts (HFF) are distantly related cell types, and therefore show many biological differences.

### Twins can be trained to be robust against artefacts resulting from discrepancies in sequencing depth

Differences in sequencing depth are a potential source of noise in the analysis of Hi-C data. To understand how differences in sequencing depth affect the training of a Twins network we employed two additional Hi-C replicates for DP thymocytes that we sequenced to twice the depth of the replicates we used for the analysis of DP and CD4 SP differentiation shown above. For simplicity, we call these DP replicates high-depth R3 and R4. We then trained a network specifically to learn the comparison between high-depth and low-depth Hi-C data with the unequal Hi-C resolutions as 'conditions' (Fig. 6a). This network was able to train efficiently, and both components of the loss decreased significantly over the course of training (Fig. 6b). The network learnt differences in Hi-C data related to sequencing depth and was able to separate DP replicates with high and low sequencing depth (Fig. 6c, d). This demonstrates that it is possible for the Twins network to learn sequencing depth-related characteristics of Hi-C data if differences in sequencing depth occur across conditions.

We compared these results with the output of our earlier T-cell differentiation network that had been trained on evenly matched replicates of from CD4 SP and DP cells. This network did not regard the two different DP resolutions (Fig. S6a). In comparison to our well-trained T-cell differentiation network, the sequencing depth network has more noisy integrated gradient maps (Fig. S6b, c). This is an indicator that the network has learnt an artefact relating to sequencing depth. This is because discrepancies in sequencing depth can change both the sparsity and the fine-scaled features observed in chromatin contact maps[33].

Finally, we trained a control network on one high-depth and low-depth replicate, and asked to compare these replicates to a second pair of high-depth and low-depth replicates (Fig. 6e). In this case, the training loss was unstable and did not meaningfully decrease (Fig. 6f), indicating that the network was unable to distinguish between the two replicate groupings. The resulting network did not have a meaningful separation between distance distributions of different sequencing depths (Fig. 6g, h). This demonstrates a robustness to sequencing depth-related artefacts. This analysis indicates that learning sequencing depth-related artefacts is avoidable, provided that differences in

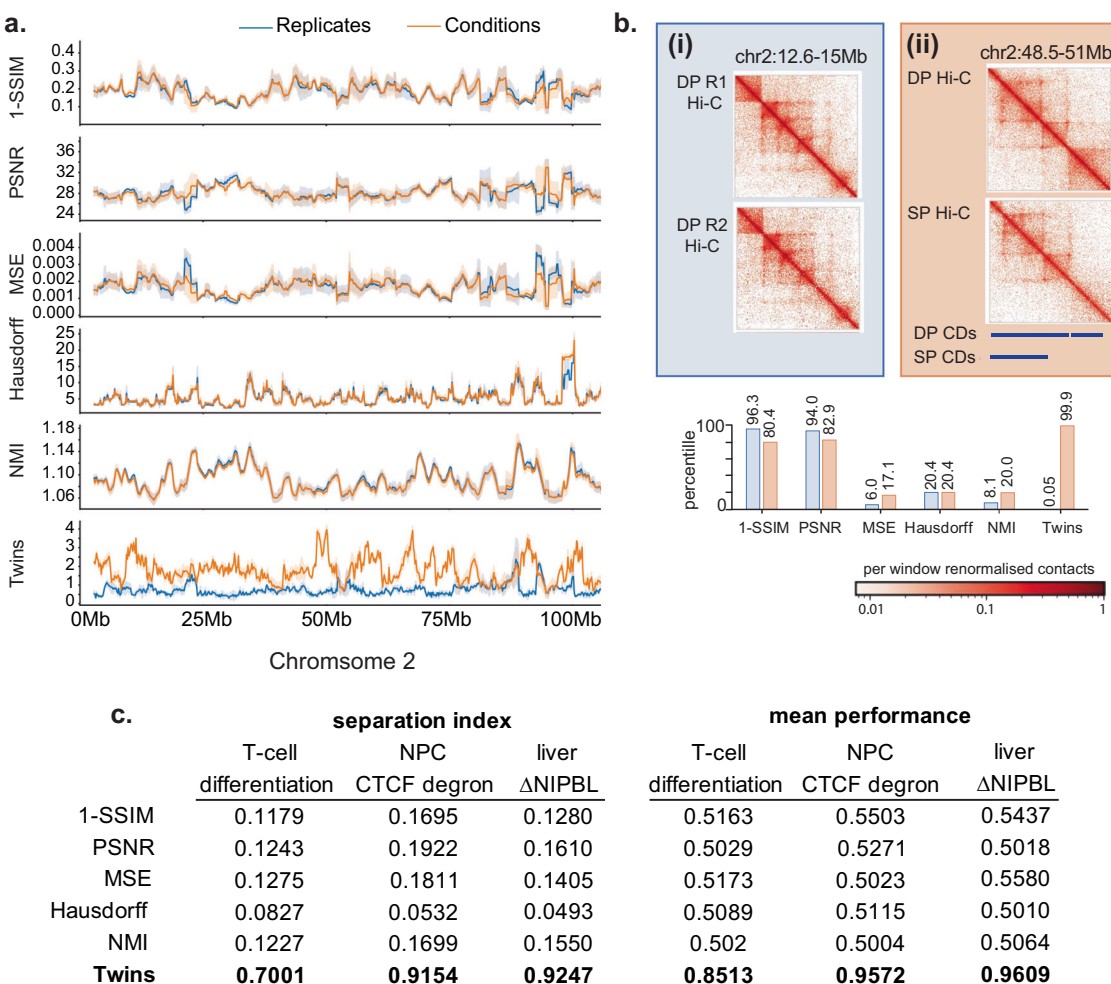

**Fig. 4 | Twins outperforms naive image similarity metrics. a** Normalised Hi-C maps at 10 kb resolution generated from DP and CD4 SP T-cells were split into regions measuring 2.56 Mb in size, and the resulting images were paired by genomic location. The distance between the pairs was calculated for 5 image similarity metrics and Twins for conditions (orange) and replicates (blue) are plotted across test chromosome 2. Data are presented as mean values ± the 95% confidence interval. For the 5 image similarity metrics there is no separation between the conditions and replicates. **b** Image similarity scores normalised by percentile across condition pairs on chromosome 2 for two example regions. (i) is a noisy replicate pair taken from DP thymocytes and (ii) is a pair from DP to CD4 SP differentiation with a visible and quantifiable loss of contact domain on the bottom right. All the naive image similarity metrics score the two regions as having a similar amount of difference or score the noisy region ahead of the visible change. **c** Quantification across the test chromosome of naive image similarity metrics and twins for all three datasets; T-cell differentiation from DP to CD4 SP, CTCF degradation in neural progenitor cells and Δ NIPBL in hepatocytes. Across all datasets using the separation index and mean performance Twins outperforms the naive image similarity metrics.

**c.**

| | separation index | | | mean performance | | |
|---|---|---|---|---|---|---|
| | T-cell differentiation | NPC CTCF degron | liver ΔNIPBL | T-cell differentiation | NPC CTCF degron | liver ΔNIPBL |
| 1-SSIM | 0.1179 | 0.1695 | 0.1280 | 0.5163 | 0.5503 | 0.5437 |
| PSNR | 0.1243 | 0.1922 | 0.1610 | 0.5029 | 0.5271 | 0.5018 |
| MSE | 0.1275 | 0.1811 | 0.1405 | 0.5173 | 0.5023 | 0.5580 |
| Hausdorff | 0.0827 | 0.0532 | 0.0493 | 0.5089 | 0.5115 | 0.5010 |
| NMI | 0.1227 | 0.1699 | 0.1550 | 0.502 | 0.5004 | 0.5064 |
| **Twins** | **0.7001** | **0.9154** | **0.9247** | **0.8513** | **0.9572** | **0.9609** |

sequencing depth are mirrored between replicates and conditions. Where mirrored data is unavailable it may be possible to minimise the effects of disparities in sequencing depth by sub-sampling the contacts from the chromatin conformation maps.

### Twins networks reveal differential Hi-C features that reflect the nature of the underlying perturbations

Based on the ability of the Twins network to distinguish meaningful differences in Hi-C from technical noise, we applied the Twins learnt convolution filters to the extraction of differential features from chromatin conformation maps. Figure 7 shows how differential feature detection can be conducted using the Twins network. We applied the convolutional layers of the trained Twins network and amalgamated the results across the genome to generate the 'Twins features' map. We then apply a threshold to this map and use a convex hull operation on the isolated features. We find that visually this gives a good approximation of the differential features visible in the Hi-C maps (Fig. 7).

To understand how different features may arise from different perturbations we applied our feature detection method to

hepatocytes comparing ΔNIPBL and tamoxifen control and also to CTCF degradation in the neural progenitor cells. For each dataset we took the isolated features, re-scaled them and, then applied a k-means clustering (k = 10) to the resulting images. We then grouped the resulting clusters manually into three broad categories; 'Domain-like' features resembling contact domains, 'Asymmetric domain-like' features which resemble part of a contact domain and 'Stripes' (Fig. 8a). We find that consistent with the role of NIPBL in loop extrusion, almost all features are lost in the ΔNIPBL, whereas some features are lost and others gained following CTCF degradation. Interestingly, stripes can be gained following degradation of CTCF, and could suggest the presence of other barriers to loop extrusion[34]. Strikingly, after loss of NIPBL most of the features affected are 'domain-like' in comparison after degradation of CTCF most are asymmetric domains or stripes, reflecting the distinct functions of cohesin and CTCF in genome organisation.

## Discussion
Current algorithms for the analysis of chromatin conformation capture excel in the quantification of specific features, but limited in their

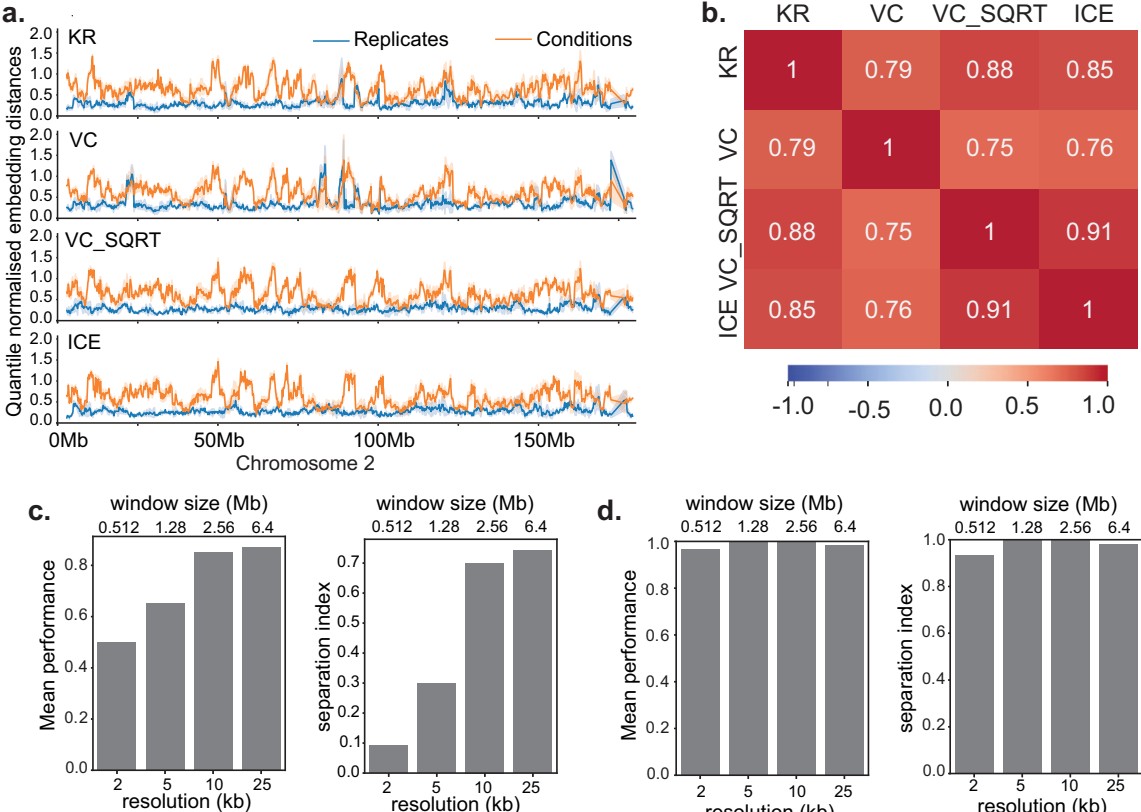

**Fig. 5 | Twins is robust to Hi-C normalisation methods and has a data set-dependent optimal operating resolution. a** Twins embedding distances on Chromosome 2 for networks trained on T-cell Hi-C maps with different normalisations. Data are presented as mean values ± the 95% confidence interval. **b** Correlation coefficient between Twins embedding distances on Chromosome 2 for networks trained and tested on T-cell Hi-C maps with different normsalisations. **c** Mean performance and separation index of Twins network for mature versus immature thymocyte Hi-C at different resolutions. Source data are provided as a Source data file. **d** Mean performance and separation index of the Twins network for Micro-C at different resolutions. Source data are provided as a Source data file.

ability to discover novel or unexpected features. This may lead to important features of chromatin conformation capture being overlooked. For example, chromatin jets have only recently been identified as Hi-C features and, once discovered, have helped to elucidate properties of cohesin-driven loop extrusion[35,36]. Advanced algorithms are therefore required to process and quantify chromatin conformation capture data in a feature-agnostic manner.

A major challenge in the analysis of chromatin conformation capture data is the presence of significant levels of non-uniform noise, which can arise for example from variations in base composition and the non-uniform distribution of restriction sites across the genome. The use of replicates to estimate and control the impact of noise is well established for the analysis of genomics data[37–39]. Although feature-agnostic analysis methods for chromatin conformation maps exist[30], these rely on naive image similarity metrics, and may be less able to effectively differentiate biological differences from technical noise than a replicate-based approach. Here, we introduce the use of Hi-C replicates for contrastive loss learning in an approach we have termed Twins. Using this approach we demonstrate that the use of replicate-based machine learning can produce informative results on chromatin conformation capture data. We show that replicate-based training can produce meaningful embedding distances in multiple biological contexts including the identification of subtle differences during T-cell development. Further, we find that using the replicates is sufficient to protect against false or exaggerated differences which could appear due to enforced training. In contrast to naive image similarity metrics, some of which have been applied to Hi-C analysis in the past[30], replicate-based training enables Twins to ignore noise and focus on differences between

biological conditions. We tested the effects of Hi-C normalisation and sequencing depth. trained on datasets of different sequencing depth, Twins can learn sequencing depth artefacts. However, if differences in sequencing depth are mirrored between conditions, the use of replicates protects from learning sequencing depth artefacts. One limitation of the image-based Twins approach is that it can not assess very long-range intrachromosomal or interchromosomal interactions.

Finally, we demonstrate that Twins can be used for reliable feature detection, and that the learnt convolutional filters are sufficient to reveal distinct features arising from perturbation of the genome organisers cohesin and CTCF. We envisage numerous other applications for Twins in the field of chromatin conformation, including the analysis of normal versus disease states in development, homeostasis, and regeneration. It is therefore important that Twins is fully compatible with the analysis of Micro-C as well as Hi-C. In conclusion, the Twins algorithm is able to produce two key meaningful outputs. The embedding distance indicates differences between conditions for chromosomal positions genome-wide and can be correlated and compared with chromatin features of interest. The extracted differential features indicate the direction of change, as well as the shape and size of features. They will serve as a useful guide for choosing suitable tools for downstream quantification of chromosome conformation contact maps.

One major outcome of this work is the processing of chromatin conformation data into a format which can be applied to other machine learning architectures to build on the approach. Leveraging of replicates by contrastive loss training could be combined with new approaches such as fully convolutional networks[40] that are emerging

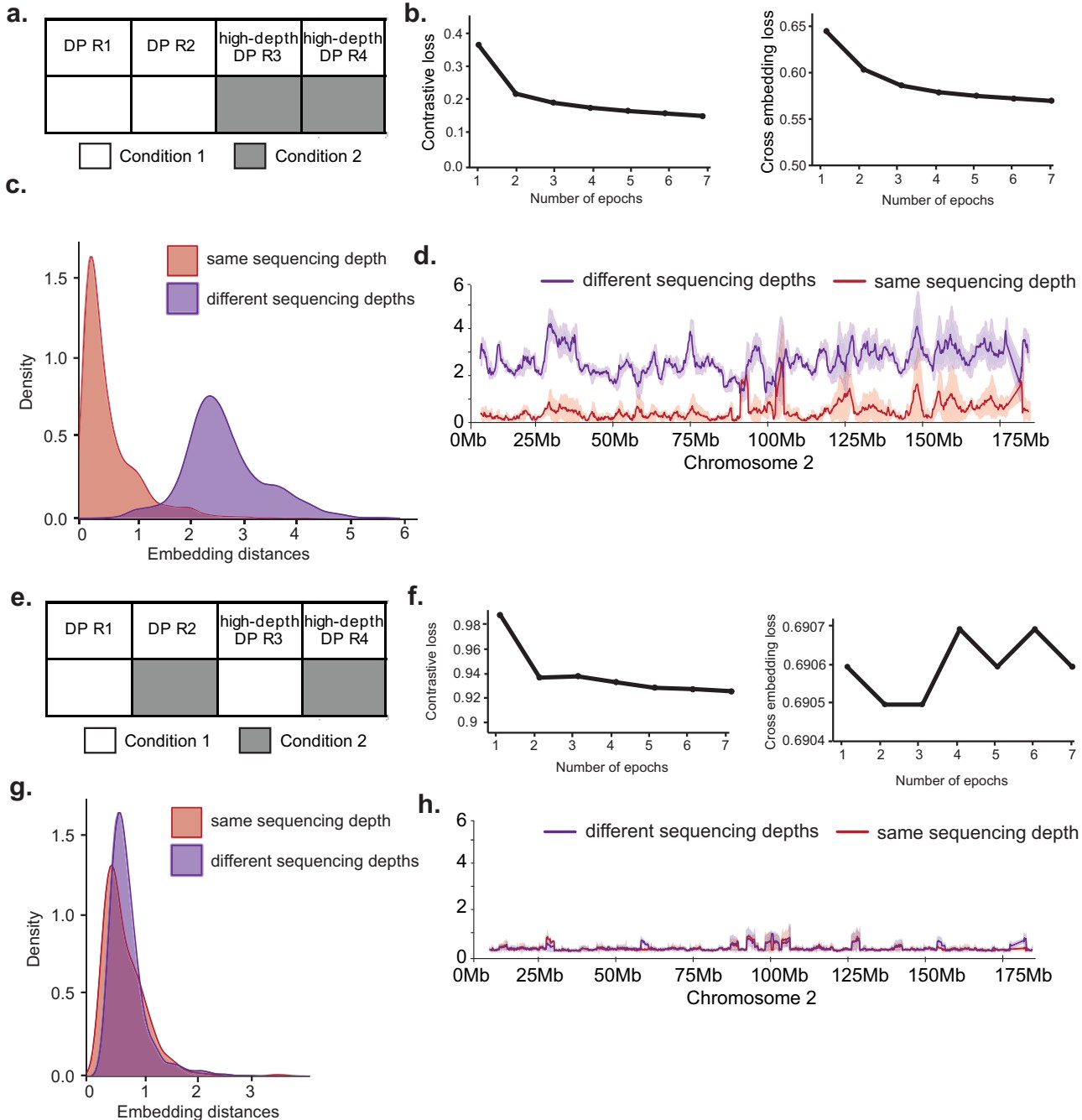

**Fig. 6 | Twins can learn to be robust discrepancies in sequencing depth.**
**a** Schema describing the training process a network which is trained to learn sequencing depth artefacts. High-resolution and low-resolution Hi-C maps are compared to another pair as though they are from separate conditions. **b** Contrastive loss and cross embedding loss terms during training of a network which follows the schema described in (**a**). **c** Twins embedding distances distribution on test chromosome for comparison between high-depth and low-depth Hi-C windows using network trained to learn sequencing depth artefacts. **d** Twins embedding distances across chromosome 2 for comparison between high-depth and low-depth Hi-C windows using network trained to learn sequencing depth

artefacts. Data are presented as mean values ± the 95% confidence interval. **e** Schema describing the training process for the control network, a network which learns to be robust to sequencing depth artefacts. One high-resolution and one low-resolution replicate are compared to another pair as though they are from separate conditions. **f** Contrastive loss and cross embedding loss terms during training of a network which follows the schema described in (**e**). Loss does not decrease significantly and the network does not learn artificial differences. **g** As in (**c**) but using network trained to be robust to artefacts. **h** As in (**d**) but using network trained to be robust to artefacts.

from the field of change detection to transform the analysis of chromatin conformation data.

## Methods
### Statistics and reproducibility
No data were excluded from the analyses.

## Data sources and processing
**ChIPseq.** Peaks from ChIPseq data were downloaded directly from the files readily available at the given sources (Table 1). For the CTCF peaks, motif orientation was detected using fimo[41] using the MA0139.1 motif. Approximately 81% of peaks contained a motif at $p < 1e - 5$.

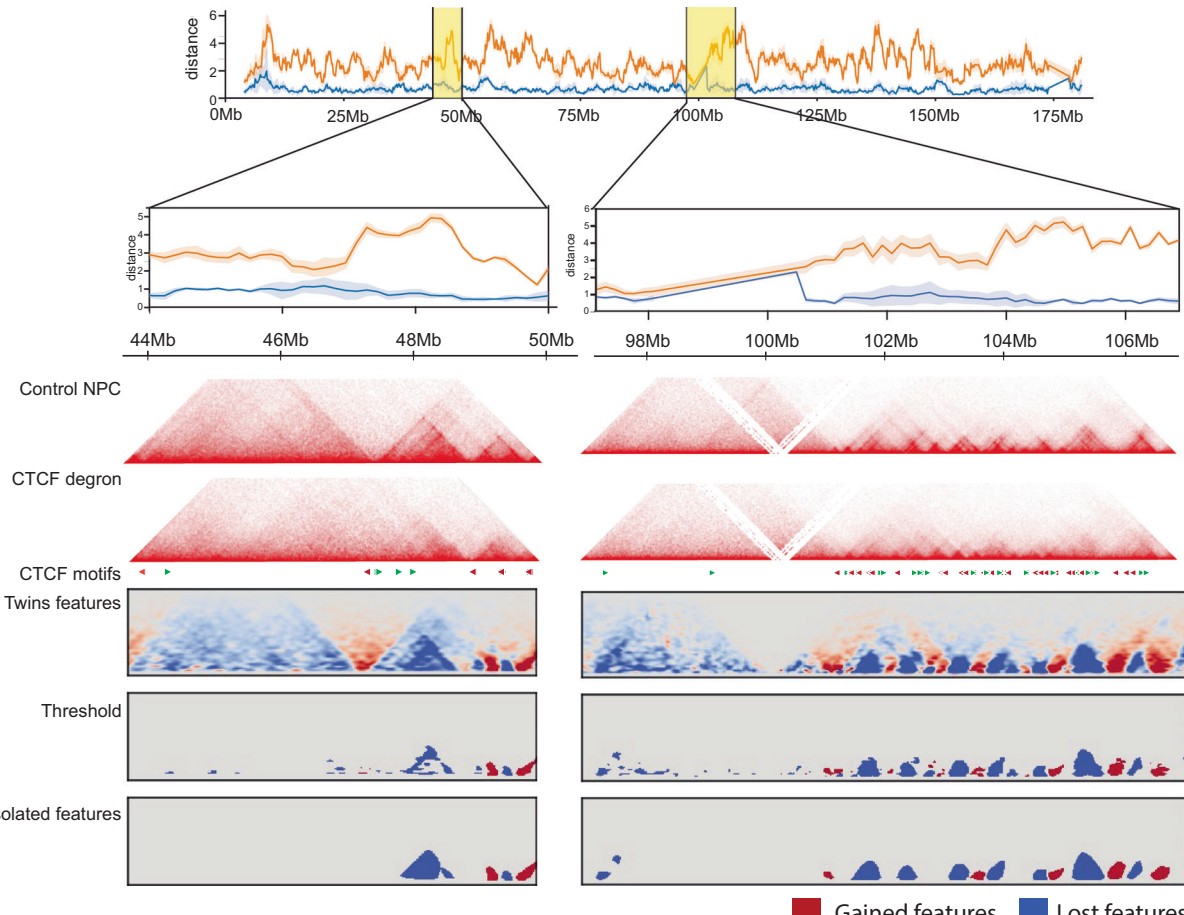

**Fig. 7 | Twins can be used for differential feature detection.** An example of two regions along chromosome 2 in neural progenitor cells before and after CTCF degradation. The Twins distance quantifies the level of change in the region here the embedding distances are presented as mean values ± the 95% confidence interval. Using this we can apply our learnt Twins filters to the Hi-C maps. These filters accentuate features of interest which can then be found by thresholding followed by isolation and the application of a convex hull transformation.

**Hi-C and Micro-C.** Where possible datasets were downloaded from publicly available files (Table 2). For all the other datasets we process the raw sequencing files using the distiller nextflow pipeline (v0.3.3) to produce .mcool files, respectively, for each experimental replicate. Raw sequencing files were downloaded using sratoolkit (v2.10.7) in fastq format, were aligned with max_mismatch_bp = 3 and then filtered by mapq ≥30. Unless explicitly stated we normalise each interaction map by either KR (if .hic) and ICE (if .mcool).

Compartment eigenvector values were determined at 100 kb resolution using[42] (v0.8.10), A and B compartments were then assigned by using the GC-content. Hi-C contact domains were called using Arrowhead with default parameters at 5 kb and 10 kb resolution with conflicts between the 10 Kb and 5 kb resolved by keeping the smaller contact domain as described in previous studies[1,43].

## Twins algorithm

**Twins data processing.** The Twin processing pipeline relies on Straw[43] (v0.0.8) and Cooler[42] (v0.8.10) modules in python in order to extract normalised windows from the interaction maps in .hic and .cool formats. Window of resolution $R$ and of size $S = 256 \times R$ are extracted from the interaction maps with a stride indicating the level of overlap between windows. Unless explicitly stated otherwise $R = 10$ kb and $S = 2.56$ Mb and the stride was 160 kb. To maintain the number of training points when the resolution/size are varied in Fig. 5, the stride is also adjusted. The parameters are available in Table S1.

We tested the effects of varying the stride. The stride impacts number of data points used at training which has three key consequences. The first consequence is that there is a higher potential for the network to over-fit on the train data. The second is that train time will be longer. Finally, the file size, as the stride becomes smaller the file size becomes larger and with multiple networks to train in a size limited environment this constraint can be a significant for most users (see Table S2).

For each window, any nan values are set to 0 and it is re-normalised by dividing by the maximum value in the window, this is to avoid underflow errors. Some windows are filtered out due to a high number of nan or 0 values, for this paper we filter any window where the number of nan values is more than the number of nonzero values and where the number positions containing with no information (i.e. all 0 values) exceeds 10% of the window. We found that this criteria was necessary to avoid training on empty data points which can lead to dead neurons in the siamese network.

These data are formatted into pytorch Datasets[44] (torch v1.6.0) and saved in separate files for each replicate/condition and for test (chromosome 2), validation (chromosome 18) and training (all other chromosomes). Before training they are paired by genomic location with label 0 if the pair of windows is from the same condition group and label 1 if the pair of windows is from different condition groups.

To fulfil the criteria of mirrored sequencing depths, our datasets are selected such that the span of the sequencing depths overlap for

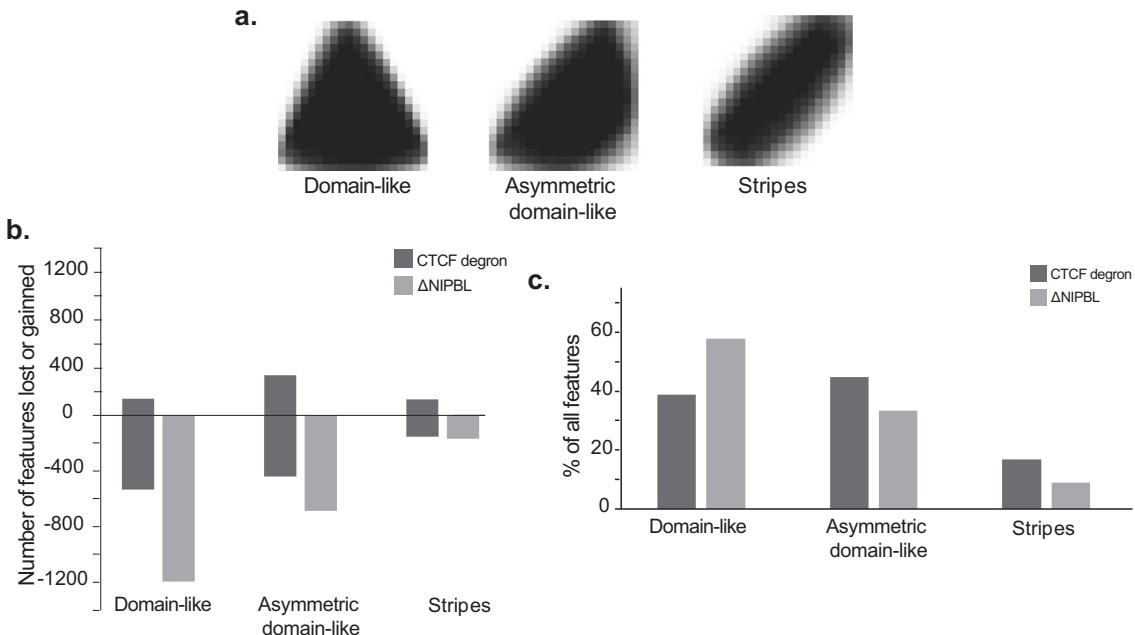

**Fig. 8 | The Twins network shows distinct differential features in the absence of NIPBL or CTCF. a** Aggregate features for the three groupings of clusters, `domain-like', `asymmetric domain-like' and `stripes'. **b** Counts of the number of differential features obtained from the ΔNIPBL and CTCF degron datasets split into three groups by cluster. Source data are provided as a Source data file. **c** Percentage of features from the different groups `domain-like', `asymmetric domain-like' and `stripes' for ΔNIPBL and CTCF degron datasets. Source data are provided as a Source data file.

**Table 1 | Information on ChIPseq peaks and their availability**

| Dataset | Organism | Conditions | Protein | No. peaks | Availability |
|---|---|---|---|---|---|
| NPCs | Mouse | Untreated control NPCS (day 4) | CTCF | 29782 | [25](Supplementary Table 2) |
| Liver cells | Mouse | Tamoxifen control | Rad21 | 44306 | GSM2740561 |
| | | | Smc3 | 61195 | GSM2740563 |

example given two replicates of sequencing depths $x_1$ and $x_2$ a network may be trained with a conditions containing two replicates of sequencing depth $y_1$ and $y_2$ and the intervals $[x_1, x_2]$, $[y_1, y_2]$ overlap (see Table S3).

**Network parameters.** We used a classical Siamese network with various adaptions. Firstly, owing to the large levels of irrelevant information/noise emitting from the diagonal of the interactions maps, we mask the diagonal. In our network we choose to remove the 3 pixels along the diagonal which is usually 30 kb ($3 \times R$). After this masking we use the standard LeNet[45] parameters however we replace the ReLU layers with Gaussian Error Linear Units (GeLU) layers[46]. This is because interaction maps are sparse and dead ReLU layers often become a problem.

In order to find the best learning rate we trained on networks with learning rate ∈ {0.001, 0.002, 0.005, 0.01, 0.02, 0.05} and random seed ∈ {30008, 72689, 50662, 60265, 30004}. We find that the learning rate has little impact on the performance so we fix the learning rate at 0.01. Final networks for all the analysis are published on GitHub.

**Twins training.** We train the twins network using the standard contrastive loss on the Euclidean distance of the embeddings ($L_1$) and add a cross embedding loss term which sits after a fully connected layer that takes the difference in the embeddings ($L_2$). This is used as a regularisation term and training with or without this term leads to highly correlated distances $p < 2.2e - 308$. We weight these such that the total loss = $2L_1 + L_2$.

In order to ensure stable training and account for the high levels of variability in the genome we use a batch size of 128 and to prevent over-fitting the network is forced to train for 5 epochs after which training is halted when the validation contrastive loss increases by more than 10%.

**Twins testing.** To test the network performance we use the Euclidean embedding distance as our learnt metric. To calculate a measure of mean performance we use the train and validation distance distributions to create a threshold for classification into replicate/condition pairs and count the rate of correct classification for replicates (replicate rate) and conditions (condition rate). The threshold is calculated using point where the replicate and condition distributions overlap. The mean performance is then the average of the replicate and condition rates. This is to take into account the in-balance between the replicate and condition distributions.

Another measure of network performance is given by separation index, this is derived from the 1 - integral of the overlap between the replicate and condition probability density distributions. This gives a metric for performance which is 1 if the replicate and condition distributions do not overlap and 0 if there is a complete overlap between the distributions.

Finally, we use the integrated gradients on individual window pairs to understand where the network places importance. To do this we deep copy the network weights into a standard CNN and then use the IntegratedGradients function from the captum package[47] to compare the windows.

**Table 2 | Information on Hi-C and Micro-C data availability and pre-processing pipelines**

| Dataset | Organism | Conditions & replicates used | Reference genome | Processing pipeline | Hi-C type | Availability |
|---|---|---|---|---|---|---|
| T-cells | Mouse | CD4SP: R1, R2 CD69negDP: R1, R2 | mm9 | HiC-Pro | .hic | GSE222211GSE199059 |
| High res T-cells | Mouse | CD69negDP: R3, R4 | mm9 | HiC-Pro | .hic | GSE199059 |
| NPCs | Mouse | auxin: R1, R2 control: R1, R2 | mm10 | | .hic | GSE94452 |
| Liver cells | Mouse | ΔNIPBL: R1, R2 tamoxifen: R1, R2 ΔNACPH2: R1, R2 | mm9 | Distiller | .mcool | GSE93431GSE122157 |
| Micro-C | Human | H1: R1, R2 HFF: R1, R3 | hg38 | Distiller | .mcool | 4DNESWST3UBH4DNES21D8SP84DNES2R6PUEK4DNESRJ8KV4Q |

**Feature extraction**. To extract features from our chromatin conformation maps we feed our windows through the convolutional layers of the network. We feed each of our windows each individually into the convolutional layers and then subtract the output and amalgamate across the genome taking the mean values across locations and filters. We then take the mean across all replicate groupings and the mean across all condition groups to produce one replicate and one condition map.

We calculate the threshold by taking the 0.95th percentile of values obtained by the application of the same method on the replicate map. We label features[48] and remove any which are smaller than 50 kb in both dimensions or greater than 2.56 Mb in size. We then apply a convex hull to our extracted features. These are re-scaled and fed as matrices into a kmeans algorithm with $k = 10$. Since the orientations are not separated, stripes pointing up or down stream are grouped into two separate clusters, similarly asymmetric contact domains are grouped according to orientations and there are several clusters which correspond to domain-like events. These clusters are grouped together by manual inspection to form the three groups discussed in Fig. 8.

### Reporting summary

Further information on research design is available in the Nature Portfolio Reporting Summary linked to this article.

## Data availability

No new data were generated as part of this study, information on data sources is available in the materials and methods. Processed data compatible with our code base and fully trained networks have been deposited in the GEO database under accession code GSE233377. Source data are provided with this paper.

## Code availability

All original code is publicly available as of the date of publication and has been version controlled on Zenodo and deposited on github[49]. Any additional code or information required to reanalyze the data reported in this publication is available from the lead contact upon request.

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

## Acknowledgements

This work was supported by the Medical Research Council UK, The Wellcome Trust (Investigator Award 099276/Z/12/Z to M.M.), EMBO (ALTF 620-2016 to Y.G.), a UK Government Industrial Strategy Rutherford Fund Fellowship (Y.G.), a UK National Productivity Investment Fund PhD studentship in Data Science or Artificial Intelligence (E.A.-J.), and the Shanghai Science and Technology Commission (20PJ1405500/21DZ2210200 to Y.G.), and ERC grant Deep4MI (884622 to D.R.).

## Author contributions

E.A.-J., Y.G., J.W.D.K., B.L., A.G.F., M.M. and D.R. conceptualised the study, E.A.-J., J.W.D.K. and M.M. analysed and visualised data, E.A.-J., M.M. and D.R. wrote the manuscript.

## Competing interests

The authors declare no competing interests.
