## [Peer Review File · Nature Communications]

A deep learning method for replicate-based analysis of chromosome conformation contacts using Siamese neural networksReviewer #1 (Remarks to the Author):

In the manuscript, the authors proposed and implemented a novel method, Twins, to generate distance via the Siamese network. Overall, the idea is reasonable and the results are great.

1, The author introduces the Siamese network into the Hi-C analysis area. The detailed implementation requires a significant amount of work to solve the difference between the Hi-C and the original Siamese application field (e.g. image processing). The authors have done solid work to overcome the challenge from data representation to the sample definition.

2, The results strongly support the hypothesis. The authors utilized a NIPBL to validate results while NIPBL is not used in the training at all. The results show the obtained embeddings are with high quality.

Revision suggestions:

1. There are some prior work on the related problem, which the authors should cite and do some comparison analysis if possible.

2. I would like the authors to further discuss the loss function in the Siamese network. Based on the manuscript, the author uses the distance-based loss function. This type of loss functions are widely used in the information retrieval field to overcome some specific engineering difficulties on a large scale (see this <https://arxiv.org/pdf/2006.11632.pdf> as a good reading material). Under Hi-C scenarios, the best use case for this loss function should be "given a genomic region, find the nearest N most similar region from the entire genome. However, the distance-based loss has significant drawbacks. The distance has no physical meanings and it wholly optimizes comparison contrast. I suggest the authors investigate using the probability based loss with some modification of the network work. The two embeddings can be concatenated, then add some fully connected layer, then use the loss function like log-loss. By using this type of the loss, the output of the network is meaningful, which is the probability that two Hi-C blocks are the same. These are suggestions and authors may deep dive and implement it for discussion.

3. This work is the foundations for many downstream research work. I suggest the author should clearly suggest the potential outputs. Based on my suggestions, there are three outcomes: 1) Using the model to generate embedding for a block of genome, which is technically a dimension reduction; 2) Finding the most similar pattern with a given genome block. 3) Compute distance between two blocks of genome (either same position from different tissues or same tissue from different position); 4) As a method to train a new model when more data (including Hi-C and the label) is available in future. The authors should give full code with documentation on all of the use cases to help non-computer science major researchers to use this work. Also, the author should also consider publishing the computed embeddings for other researchers to use without installing any deep learning tools.

Reviewer #2 (Remarks to the Author):

This manuscript proposed a deep learning method, Twins, to distinguish biological variation from technical noise and extract biological meaning features. The results look promising and well done. They proposed Twins as a powerful tool for the exploration of chromosome conformation capture data, such as Hi-C, capture Hi-C and Micro-C. As a tool development manuscript, the potential impact is its application in other studies. It said in the method that "All original code has been deposited on Git Hub and is publicly available as of the date of publication." Since I couldn't see the code and related instructions, I couldn't evaluate how easy for other researchers in the community to apply their method to their research.

Major criticism:

1. In Figure 1C and 1D, shows that embedding distances can separate biological differences from

technical noise, which is very promising. However, Figure 6C shows that a 2-fold sequencing depth difference can also make such separation. Though Figure 6G justifies that if differences in sequencing depth are mirrored between replicates and conditions, the sequencing depth-related artifacts are avoidable, it's hard to have exact mirrored sequencing depth in the real world. It will be helpful if the author can provide some practical strategies to deal with the data without the same/mirrored sequencing depth.

2. It was claimed that the method applied for the normalization of Hi-C data does not impact Twins' performance based on the visualization of distance distribution across chromosome 2 and the correlation coefficient of all normalization methods. However, Figure 5A shows that distance based on VC normalization is very different from the other three methods, even though the correlation coefficient is not bad ($=0.79$). It will be helpful to directly show the performance of Twins based on four different normalization methods.

3. Although Twins can distinguish biological variation from technical noise, it's unclear how to utilize Twins to get differential 3D genomic architecture between two conditions. For example, it would be interesting to know which enhancer-promoter looping is changed during certain conditions.

Minor criticism:

1. Why Twins perform much better for Micro-C than Hi-C, especially in 2-5kb resolution as shown in Figure 5C and 5D. It should be discussed.
2. Figure 2E, it's hard to tell the difference between control NPC and CTCF degron.
3. A typo in Figures 2C and 2D. It should be 'As in (B)...'
4. Is there a particular reason to choose chromosome 2 as a test chromosome?
5. On page 13, a typo in the sentence "we trained a control network on one high-res and low-res replicate and asked to compare these replicates to a second pair of high-res and low-res replicates 6E). In this case, the training loss was unstable and did not meaningfully decrease 6F)...". On page 21, Figure S3A, there are two 'high'. The author should carefully check through the manuscript to minimize typos.

Reviewer #3 (Remarks to the Author):

In this manuscript, Al-jiburry et al developed a method named Twins to explore chromosome conformation capture data. The major advance of the method is to utilize biological or technical replications to distinguish technical noise from biological variations using contrastive learning. In the manuscript, the authors showed that Twins outperformed multiple naïve image similarity methods like 1-ssim. Using Twins, the authors also detected chromosome conformation features that had differential characters Overall, this work is of interest to the field of Hi-C analysis but with certain limitations. There are multiple concerns that need to be addressed before the manuscript is considered for publication.

Major comments:

1. In this work, by selecting overlapping matrixes with variable sizes from kilobases to a few million bases along the diagonal, long-range interactions and interchromosomal interactions beyond the matrix sizes were excluded, which is questionable. Recent works have shown that long-range interactions over several Mb were important for cellular function (Beagrie et al, Nature, 2017). Interchromosomal interactions were also shown to be critical for cell function (Xiong et al, Nature Communications, 2019). Technically, the ratio between inter- and intra-chromosome interactions is also important to evaluate Hi-C experiment quality. With these considerations, I would suggest the authors refine the overlapping images to include these additional interactions in their strategy. For instance, the images are not necessarily to be square.
2. The authors have analyzed multiple published datasets to demonstrate the power of Twins. However, since the chromosome conformation changes upon CTCF degradation or NIPBL deletion were genome-wide and dramatic, it was not a surprise to find large-scale differences. On the other hand, the authors observed very dramatic chromosome-wide distances between the two states of T-cell differentiation, which was quite surprising as differentiation usually need part of the genome but not the whole genome to be regulated. These observations raise some questions: 1) what's

the biological function of the distances between DP and SP? Were they all functional for T-cell differentiation? 2) If apply Twins to a condition that only a limited part of the chromosome has conformation changes, whether Twins can identify these small-scale differences? This question needs to be very well demonstrated since these small-scale differences are the most common situations in analyzing the Hi-C dataset.

3. The biological meanings of the features extracted by Twins were not very well illustrated. Although the authors showed some enrichment of CTCF, cohesion, and chromatin modifications in the different regions identified with Twins, how these differences correlate to cellular function and previously identified conformation features like TADs and loops had not been carefully examined. For instance, whether the degree of embedding distances has specific indications for chromosome conformation? Could the author add some analysis to check how the embedding distances related to TAD insulation, compartment, and CTCF/cohesin binding in Figure 2B-2D? What are the genes being affected in these regions and whether they are critical for T-cell differentiation? Most important, in Figures 7 and 8, what are the biological meanings of the differential features? Did the authors find some differential features in T-cell differentiation system?

4. Another concern about Twins is the use of parameters for the image. Authors should provide certain rationales for the use of the parameters such as why the stride was used as 160 kb in a window size of 256xR? Whether the parameters will affect the performance? How robust the performance is? If possible, the authors should provide a guideline for choosing optimal parameters.

5. In Figure 6A-D, the authors claimed that Twins could learn sequencing depth-related characteristics of Hi-C data if differences in sequencing depth occur across conditions, which was quite confusing since we would expect no conformation differences if two datasets were only different in sequencing depth. Could the authors illustrate what are the sequencing depth-related characteristics? How do these characteristics differentiate from sequencing depth-independent characteristics? Overall, Figure 6 was not clear. Further explanations are needed.

6. All the validations were performed on chromosome 2 across the manuscript. The authors should provide at least one additional validation to show another chromosome. This is important if the conformation changes are chromosome-specific.

Minor comments:

1. In Figure 2B-2D, the distances around telomeric regions are very striking. Is it a common feature or a specific feature either related to specific conditions or specific chromosomes?
2. Figure 2A: the embedding distance plots didn't have a scale.
3. Figure 4B: how the contact domains were defined? If the differential CDs were identified by another method, how did the distances identified by Twins coordinate with the differential CDs? It's would be better to include embedding distance plots for the selected regions.
4. Figures S3 and 6: please consider using "high-depth" and "low-depth" other than "high-res" and "low-res", which were not appropriate since the resolution was the same.
5. Figures 7 and 8: What's the meaning of the red and blue regions in "isolated features" in Figure 7? What are the algorithms to obtain gained and lost features? How the "threshold" was defined?
6. For Hi-C contact images in the manuscript (Figure S2C, S2D, S3B, S3C, S4B), please plot a density bar to show the contact numbers in the images.

Response to referees' comments NCOMMS-22-51138

'Twins: A deep learning method for replicate-based conformation contact map analysis',

We thank the referees for their thoughtful and constructive comments. We have performed additional work to strengthen the manuscript and to demonstrate the practical utility of the Twins approach to the broader genomics community.

In particular we have

- i) demonstrated that Twins is able to detect small-scale chromatin changes in response to local genome perturbations
- ii) documented how Twins scores relate to Hi-C changes during T cell development in relation to Hi-C features quantified by independent approaches
- iii) added use cases with documentation for the non-specialist reader in addition to information to aid with the selection of parameters
- iv) added a test dataset and tutorial for the non-specialist reader. These data and the fully trained models are available from GEO under accession number GSE GSE233377 (reviewer token glipmwykzlmfdqp).
- v) extended our comparison of Twins with naïve image similarity metrics
- vi) addressed concerns about the availability of chromosome conformation capture data with matched sequencing depth

We thank the referees for encouraging us to implement these changes, which we think have significantly strengthened the manuscript and demonstrated the practical utility of Twins to the broader genomics community.

A detailed point-by-point response to the referees' comments follows below:

Reviewer #1:

In the manuscript, the authors proposed and implemented a novel method, Twins, to generate distance via the Siamese network. Overall, the idea is reasonable and the results are great.

1, The author introduces the Siamese network into the Hi-C analysis area. The detailed implementation requires a significant amount of work to solve the difference between the Hi-C and the original Siamese application field (e.g. image processing). The authors have done solid work to overcome the challenge from data representation to the sample definition.

2, The results strongly support the hypothesis. The authors utilized a NIPBL to validate results while NIPBL is not used in the training at all. The results show the obtained embeddings are with high quality.

We thank the referee for these positive comments, and in particular their appreciation of the work we have undertaken to adapt the Siamese network approach to Hi-C applications.

Revision suggestions:

1. There are some prior work on the related problem, which the authors should cite and do some comparison analysis if possible.

We agree with the referee that it is important to place our study in the context of prior work on the differential analysis of chromatin conformation maps. Whereas Twins is designed to be feature-agnostic, most prior metrics were engineered to search for and quantify pre-defined features within chromatin conformation maps (reviewed by Gunsalus et al 2023, DOI 10.1101/2023.04.04.535480). A feature-agnostic method was proposed by Galan et al.,

2020 (DOI 10.1038/s41588-020-00712-y). Their approach was to combine two naïve image similarity metrics, SSIM and PSNR. In contrast to Twins, this method does not utilise replicates in order to distinguish technical from biological variation, and there has been debate over whether or not SSIM is able to resolve technical noise from true differences in chromatin conformation (Ing-Simmons et al 2021 DOI, 10.1101/2021.10.18.464422; Lee et al., 2021 DOI, 10.1101/2021.09.23.459925). Consequently, we have compared Twins with both SSIM and PSNR individually on per-window scaled Hi-C images in Fig. 4. The results demonstrate that Twins is able to better resolve genuine differences in chromatin conformation. In further support of this conclusion, we have also computed two additional comparison metrics, F1 score and area under the receiver operator curve (AUROC) summarised in Referee 1 Table 1. We have added the following sentence to the discussion of our revised manuscript: 'Although feature-agnostic analysis methods for chromatin conformation maps exist, these rely on naive image similarity metrics, and may be less able to effectively differentiate biological differences from technical noise than a replicate-based approach.'

F1 score	Tcell	CTCF degron	NIPBL	AUROC	Tcell	CTCF degron	NIPBL
1-SSIM	0.6669	0.6668	0.6666	1-SSIM	0.5133	0.5410	0.5498
PSNR	0.6666	0.6671	0.6666	PSNR	0.4890	0.5218	0.4292
MSE	0.667	0.6666	0.6668	MSE	0.5108	0.4780	0.5701
Hausdorff	0.6663	0.6632	0.6666	Hausdorff	0.5070	0.4934	0.4829
NMI	0.6669	0.6667	0.6666	NMI	0.4792	0.4300	0.4849
Twins	0.8515	0.9586	0.9609	Twins	0.9128	0.9917	0.9927

Referee 1 Table 1. Comparison of F1 scores and area under the receiver operator curve (AUROC) between Twins and the naive image similarity metrics 1-SSIM, PSNR, MSE, Hausdorff, and NMI.

2. I would like the authors to further discuss the loss function in the Siamese network. Based on the manuscript, the author uses the distance-based loss function. This type of loss functions are widely used in the information retrieval field to overcome some specific engineering difficulties on a large scale (see this <https://arxiv.org/pdf/2006.11632.pdf> as a good reading material). Under Hi-C scenarios, the best use case for this loss function should be “given a genomic region, find the nearest N most similar region from the entire genome. However, the distance-based loss has significant drawbacks. The distance has no physical meanings and it wholly optimizes comparison contrast. I suggest the authors investigate using the probability based loss with some modification of the network work. The two embeddings can be concatenated, then add some fully connected layer, then use the loss function like log-loss. By using this type of the loss, the output of the network is meaningful, which is the probability that two Hi-C blocks are the same. These are suggestions and authors may deep dive and implement it for discussion.

We thank the referee for this suggestion. The original Twins training process included a cross entropy regularisation term at the end of a fully connected layer so that the total loss is a scaled sum of (i) the cross entropy loss at the end of the fully connected layer and (ii) the contrastive loss at the embedding layer.

In response to the referee's comment we have performed additional analyses, in which we have systematically varied the scaling factor λ of the cross entropy loss and the contrastive loss. Referee 1 Fig. 1 shows the results across chromosome 2. The results indicate that when increasing weight is placed on the cross entropy loss, the embedding distances become less meaningful, and the certainty of replicate classification becomes discrete.

Since we prefer a quantification of biological differences over certainty estimates, we feel that the contrastive loss with the embedding distance is a good solution for the problem we are trying to address in our manuscript. However, increased emphasis on cross entropy loss may be the best approach for other use cases, and as a result we have included documentation for this scenario in our code-base at github.com/ea409/twins_hic.

Referee 1 Figure 1. The effect of varying the scaling factor, λ on the replicate classification certainty and the embedding distance. As more emphasis is placed on the cross-entropy loss, the embedding distance becomes less well separated.

3. This work is the foundations for many downstream research work. I suggest the author should clearly suggest the potential outputs. Based on my suggestions, there are three outcomes: 1) Using the model to generate embedding for a block of genome, which is technically a dimension reduction; 2) Finding the most similar pattern with a given genome block. 3) Compute distance between two blocks of genome (either same position from different tissues or same tissue from different position); 4) As a method to train a new model when more data (including Hi-C and the label) is available in future. The authors should give full code with documentation on all of the use cases to help non-computer science major researchers to use this work. Also, the author should also consider publishing the computed embeddings for other researchers to use without installing any deep learning tools.

We thank the referee for suggesting these use cases and agree that documentation for these use cases will make our work more accessible to non-specialists. We have now included and documented all 4 use cases in our published code base. We note that use case 2 produced interesting results (Referee 1 Fig. 2). We have also included the computed embeddings alongside our processed data to increase the utility for researchers who are less familiar with deep learning libraries.

Referee 1 Figure 2: Illustration of use case 2 suggested by the referee - finding the most similar pattern within a given genome block. The query Hi-C image was taken from chromosome 9 35.84Mb-38.4Mb of double positive thymocytes (left) and used to search for the closest embedding vectors across all double positive thymocyte Hi-C regions. These are given by the regions at chr3: 77.92-80.48Mb and chr4: 52.16-54.72Mb.

Reviewer #2:

This manuscript proposed a deep learning method, Twins, to distinguish biological variation from technical noise and extract biological meaning features. The results look promising and well done. They proposed Twins as a powerful tool for the exploration of chromosome conformation capture data, such as Hi-C, capture Hi-C and Micro-C. As a tool development manuscript, the potential impact is its application in other studies.

We thank the referee for these positive comments.

It said in the method that “All original code has been deposited on Git Hub and is publicly available as of the date of publication.” Since I couldn’t see the code and related instructions, I couldn’t evaluate how easy for other researchers in the community to apply their method to their research.

We apologise that the referee was unable to access our code. The revised manuscript now clearly states that the GitHub code can be found here github.com/ea409/twins_hic. The supplementary data including fully trained models are available to download from GEO under accession number GSE GSE233377. The reviewer token is [glimpykzlmfdqp](https://www.ncbi.nlm.nih.gov/geo/query/acc.cgi?acc=GSE233377). In addition, during revision we have added a test dataset and tutorial on our GitHub in order to enhance the practical utility of our approach to the broader genomics community. This code will be saved and version-controlled on Zenodo once the review process is complete and the code base is stable.

Major criticism:

1. In Figure 1C and 1D, shows that embedding distances can separate biological differences from technical noise, which is very promising. However, Figure 6C shows that a 2-fold sequencing depth difference can also make such separation. Though Figure 6G justifies that if differences in sequencing depth are mirrored between replicates and conditions, the sequencing depth-related artifacts are avoidable, it’s hard to have exact mirrored

sequencing depth in the real world. It will be helpful if the author can provide some practical strategies to deal with the data without the same/mirrored sequencing depth.

We appreciate the referee's concern that real-world chromosome conformation capture data will rarely be perfectly matched for sequencing depth. Our experience with the data sets used in our study (Referee 2 Table 1 and Table 5 in the revised manuscript) indicates that differences in sequencing depth can be well tolerated so long as there as the span of the sequencing depths overlap for example given two replicates of sequencing depths x_1 and x_2 a network may be trained with a conditions containing two replicates of sequencing depth y_1 and y_2 and the intervals $[x_1, x_2]$, $[y_1, y_2]$ overlap. To clarify this point we have included the follow statement in the materials and methods “To fulfil the criteria of mirrored sequencing depths, our datasets are selected such that the span of the sequencing depths overlap for example given two replicates of sequencing depths x_1 and x_2 a network may be trained with a conditions containing two replicates of sequencing depth y_1 and y_2 and the intervals $[x_1, x_2]$, $[y_1, y_2]$ overlap”.

T-cell differentiation		
	R1	R2
DP	244438962	223193977
SP	222906061	209206741
DP high depth	417830413	427257794
downsampled	208915207	213628897

Nipbl deletion		
	R1	R2
TAM	51998355	75626897
WT	76277577	65903921
NCAP2	67503333	63246470
NIPBL	65081892	72757890

CTCF degran		
	R1	R2
Control	845858980	838742819
Auxin	737448939	843972188

Micro-C		
	R1	R2
H1	1470519840	1749983591
HFF	3049701785	1508233177

Referee 2 Table 1. Sequencing depth of the data sets used in our study.

Where matched chromosome conformation capture data are not available, subsampling at the level of Hi-C contact maps or at the read level can be used, but may not give perfect results (Referee 2 Fig. 1). One way in which a user could circumvent this issue is by subsampling the data in different ways and including all of these in training. We have now detailed this in our code documentation and by including the table of sequencing depths and the following sentence in the results “Where mirrored data is unavailable it may be possible to minimise the effects of disparities in sequencing depth by sub-sampling the contacts from the chromatin conformation maps.”

Referee 2 Figure 1. Twins embedding distances on subsampled data for a network trained to look at sequencing depth differences and a network robust to sequencing depth differences. Subsampling Hi-C data is not always sufficient to reduce biases related to sequencing depth.

2. It was claimed that the method applied for the normalization of Hi-C data does not impact Twins' performance based on the visualization of distance distribution across chromosome 2 and the correlation coefficient of all normalization methods. However, Figure 5A shows that distance based on VC normalization is very different from the other three methods, even though the correlation coefficient is not bad ($=0.79$). It will be helpful to directly show the performance of Twins based on four different normalization methods.

We thank the referee for this observation. In order to more directly show the performance of twins based on the normalisation methods we have included a version of Fig. 5A with the axis scaled by quantile. Here a score of 1 demonstrates that a value is in the 0.95 quantile (Referee 2 Fig. 2, new Fig. 5A of the revised manuscript). As is now evident, even in the VC normalised Hi-C maps, the embedding distances do still follow the same distribution across chromosome 2. To illustrate this point further, we have superimposed each replicate and condition distribution scaled by quantile (Referee 2 Fig. 3).

Referee 2 Figure 2 (new Fig. 5A of the revised manuscript) Twins embedding distances for T-cell differentiation networks across chromosome 2 for different Hi-C normalisations scaled by 0.95 quantile.

Referee 2 Figure 3. Embedding distance for Twins networks with different normalisations scaled to fit the same axis and overlaid.

3. Although Twins can distinguish biological variation from technical noise, it's unclear how to utilize Twins to get differential 3D genomic architecture between two conditions. For example, it would be interesting to know which enhancer-promoter looping is changed during certain conditions.

Twins produces two outputs that will be useful for delineating differences in genome architecture between conditions, the embedding distance and Twins features. The utility of each output is discussed below.

The Twins embedding distance indicates differences between conditions for chromosomal positions across the genome. In Figures 2E-F of the revised manuscript we illustrate how the Twins embedding distance can be useful in identifying relationships between chromatin features and changes in chromosome conformation contact maps.

We find that the density of cohesin peaks strongly correlates with the Twins embedding distance for Nipbl ko, and the density of CTCF peaks strongly correlates with the Twins embedding distance for CTCF degnon. To illustrate specificity, we have added an analysis of ChIP-seq signal for the histone modification H3K27me3 in the CTCF degnon system to Figure 2H of the revised manuscript (shown below as Referee 2 Figure 4). The results show that H3K27me3 is not predictive of Twins embedding distances. To make these analyses more visually intuitive we have added genome browser shots of Twins embedding distances versus the density of CTCF ChIP-seq peaks, contact domains and deregulated genes (Supplementary Figure 2 of the revised manuscript, shown below as Referee 2 Figure 5). Taken together, these data illustrate how Twins embedding distances can be used to identify chromatin features that correlate with - and may be causative for - changes in chromosome conformation contact maps identified by Twins embedding distances.

Referee 2 Figure 4 (New Fig. 2F of the revised manuscript). In contrast to the density of CTCF peaks, H3K27me3 is not predictive of Twins embedding distances in CTCF-depleted neuronal progenitor cell ($p > 0.05$).

Referee 2 Figure 5 (New Supplementary Fig. 2 of the revised manuscript). Regions with high Twins scores after CTCF depletion in neuronal progenitor cells are characterised not only by a high density of CTCF binding, but also by a high density of CTCF motifs and changes in contact domains, as well as the presence of deregulated genes.

In addition, as demonstrated by new data shown in Fig. 2G of the revised manuscript, the embedding distance is sensitive to small scale changes in chromatin conformation, and therefore able to detect local genomic perturbations.

New Fig. 2G of the revised manuscript. Twins scores reflect focal perturbations in chromosome architecture.

Isoda et al., 2017 deleted a distal enhancer of the developmentally regulated *Bcl11b* gene on chromosome 12. The location of the enhancer at 108.4Mb and of the *Bcl11b* gene are shown. Enhancer deletion results in localised changes in the Hi-C map. Twins scores across chromosome 12 show a prominent peak at the location of the *Bcl11b* enhancer.

2) Twins features indicate the direction of change and the shape as well as the size of features identified (Fig. 7 of the revised manuscript).

The direction of change, the shape, and the size of features identified by Twins will serve as a useful guide for choosing suitable tools for downstream quantification of chromosome conformation contact maps. For example, feature size will indicate whether changes preferentially occur on the scale of TADs or contact domains (Referee 2 Fig. 5). Feature shape will indicate whether changes are symmetrical, asymmetric, or even stripe-like (Fig. 8 of the revised manuscript). This will facilitate the use of suitable downstream analysis tools.

To increase the practical utility of Twins to the broader genomics community we have added the following text to the discussion “In conclusion, our Twins algorithm is able to produce two key meaningful outputs. The Embedding distance which indicates differences between conditions for chromosomal positions genome-wide which may be correlated and compared with chromatin features of interest such as protein binding. The extracted differential features which indicate the direction of change and the shape as well as the size of features which will serve a useful guide for choosing suitable tools for downstream quantification of chromosome conformation contact maps”.

Minor criticism:

1. Why Twins perform much better for Micro-C than Hi-C, especially in 2-5kb resolution as shown in Figure 5C and 5D. It should be discussed.

We agree with the referee that Twins performs much better for Micro-C than Hi-C. We think that there are technical as well as biological reasons for this, as explained in the following paragraph that we have added to the discussion of the revised manuscript: “Twins performed noticeably better on Micro-C than Hi-C data. There are several possible reasons for this. Micro-C has higher resolution, and therefore contains many more fine-scaled contacts. The increased resolution of Micro-C also enables near-optimal performance of the Twins network across a range of window sizes and resolutions (Fig. 5D). In addition, human embryonic stem cells (H1) and fibroblasts (HFF) are distantly related cell types, and therefore show many biological differences”.

2. Figure 2E, it's hard to tell the difference between control NPC and CTCF degran. We appreciate the feedback and have adjusted the layout of the figure accordingly.

3. A typo in Figures 2C and 2D. It should be ‘As in (B)...’
We apologise for this mistake and thank the referee for pointing it out.

4. Is there a particular reason to choose chromosome 2 as a test chromosome?
In our paper we have kept chromosome 2 as the test chromosome to maintain consistency in the test dataset size. However, there is no particular reason and in response to this and a similar comment by referee 3 we have generated supplementary figures which demonstrate that the use of other chromosomes as a test leads to very highly correlated results.

5. On page 13, a typo in the sentence “we trained a control network on one high-res and low-res replicate and asked to compare these replicates to a second pair of high-res and low-res replicates 6E). In this case, the training loss was unstable and did not meaningfully decrease 6F)...”. On page 21, Figure S3A, there are two ‘high’. The author should carefully check through the manuscript to minimize typos.
We apologise to the reviewer for the typos and have amended the manuscript accordingly.

Reviewer #3:

In this manuscript, Al-Jibury et al developed a method named Twins to explore chromosome conformation capture data. The major advance of the method is to utilize biological or technical replications to distinguish technical noise from biological variations using contrastive learning. In the manuscript, the authors showed that Twins outperformed multiple naïve image similarity methods like 1-ssim. Using Twins, the authors also detected chromosome conformation features that had differential characters.

We thank the referee for these positive comments.

Overall, this work is of interest to the field of Hi-C analysis but with certain limitations. There are multiple concerns that need to be addressed before the manuscript is considered for publication.

Major comments:

1. In this work, by selecting overlapping matrixes with variable sizes from kilobases to a few million bases along the diagonal, long-range interactions and interchromosomal interactions beyond the matrix sizes were excluded, which is questionable. Recent works have shown that long-range interactions over several Mb were important for cellular function (Beagrie et al, Nature, 2017). Interchromosomal interactions were also shown to be critical for cell function (Xiong et al, Nature Communications, 2019). Technically, the ratio between inter- and intra-chromosome interactions is also important to evaluate Hi-C experiment quality. With these considerations, I would suggest the authors refine the overlapping images to include these additional interactions in their strategy. For instance, the images are not necessarily to be square.

We thank the referee for this suggestion. We agree with the referee that interchromosomal interactions can be extremely important. However, these types of features are not readily accessible to the Twins algorithm, as they may not form features in the computer vision sense. We now acknowledge this limitation in the discussion of the revised manuscript “One limitation of our approach is that it is not possible to assess changes beyond the diagonal of chromatin conformation maps where there may be other features of interest.” We opted to use squares of the dimension 256x256 because it has many curated network architectures; finding good network architectures is non-trivial task, and an active area of research in deep learning.

2. The authors have analyzed multiple published datasets to demonstrate the power of Twins. However, since the chromosome conformation changes upon CTCF degradation or NIPBL deletion were genome-wide and dramatic, it was not a surprise to find large-scale differences. On the other hand, the authors observed very dramatic chromosome-wide distances between the two states of T-cell differentiation, which was quite surprising as differentiation usually need part of the genome but not the whole genome to be regulated. These observations raise some questions: 1) what’s the biological function of the distances between DP and SP? Were they all functional for T-cell differentiation? 2) If apply Twins to a condition that only a limited part of the chromosome has conformation changes, whether Twins can identify these small-scale differences? This question needs to be very well demonstrated since these small-scale differences are the most common situations in analyzing the Hi-C dataset.

We thank the reviewer for posing these important questions.

In 1), the referee questions the biological significance of Twins distances between DP and SP thymocytes. We have performed extensive analyses in response to this question,

because similar concerns are raised by this referee in their point 3 below. The data presented here also serve to address point 3 below.

New Supplementary Figure 3 of the revised manuscript (shown below) provides a visual demonstration how Twins scores correspond to Hi-C maps of 4 representative genomic regions with high Twins scores and 2 representative genomic regions with low Twins scores (Supplementary Figure 3A). We then present a formal analysis of the relationship between Twins scores and Hi-C features in T cell development. Importantly, Supplementary Figure 3B of the revised manuscript provides a link between Twins outputs and Hi-C features in T cell development. Here, we analysed changes in directionality index, insulation score, gain or loss of contact domains, and A/B compartment identity. We then classified genomic regions by the extent of change in these Hi-C features. Regions with low, medium, or high changes in Hi-C features showed significant differences in their Twins scores. For each Hi-C feature examined, Twins scores reflect the degree of change during T cell development.

New Supplementary Figure 3 of the revised manuscript. Features of genomic regions with high versus low Twins scores in T cell development.

A. Hi-C maps of 4 representative genomic regions with high Twins scores and 2 representative genomic regions with low Twins scores.

B. Twins scores reflect changes in Hi-C features during T cell development. Comparison of Twins embedding distance to changes in directionality, insulation, differential contact domains and changes in compartment identity. Mean changes in directionality at domain boundaries, total insulation and changes in compartment are quantified for each window and placed into three equal sized bins (low, medium and high) by percentile. Number of differential contact domains are counted for each window and placed into bins ([1-10], [11-

18], [19-27], [28-35] and 36+) each bin has a variable number of regions (5786, 2475, 672, 113, 19). All p-values are assigned using a two-sided t-test.

To further illustrate how Twins output relates to T cell development, Referee 3 Fig. 1 depicts Twins scores, differential ATAC-seq peaks (ATAC-seq data from Miyazaki et al., 2020 DOI: 10.1126/sciimmunol.abb1455), and contact domains in DP and CD4 SP thymocytes for the region harbouring the developmentally regulated *Rag1* and *Rag2* genes on chromosome 2. High Twins scores correspond to regions with a high density of differential ATAC-seq peaks, and gain or loss of contact domains during T cell differentiation.

Referee 3 Figure 1. Twins scores reflect developmental processes during T cell differentiation. Twins scores, differential ATAC-seq peaks (blue: downregulated between DP and CD4 SP thymocytes, red: upregulated between DP and CD4 SP thymocytes), and contact domains in DP and CD4 SP thymocytes are shown for the region harbouring the developmentally regulated *Rag1* and -2 genes on chromosome 2.

Figure 4Bii of the revised manuscript shows an additional example of a region with a high Twins score. This region experiences the loss of a contact domain that harbours the developmentally regulated *Mmadhc* gene.

Of note, changes in embedding distances may give an impression that differences are spread out over large genomic areas. Due to the use of overlapping windows, a change observed in one location has consequences on the score upstream and downstream of that location (Referee 3 Fig. 2).

Referee 3 Figure 2. Illustration of how features location can impact twins score due to overlapping windows. Briefly a feature e.g. contact domain impacts windows either side of

the central window containing the feature since they too may fully or partially contain the feature. This can make the embedding distance across the test chromosome look higher than may be expected.

Finally, we performed stringent validation to ensure that Twins does not learn false differences. Firstly, we analysed NCAPH2 ko, which had been found not to result in Hi-C changes in quiescent liver cells (Figure 3B, C). Secondly, we used random shuffling of reads (Supplementary Figure 4). In each case, we found that the Twins algorithm was robust to learning false differences in Hi-C maps.

In question 2), the referee challenges us to demonstrate whether Twins is able to identify small-scale differences that result from local experimental perturbations. To address this question we examined Hi-C data published by Isoda et al., 2017 DOI: 10.1016/j.cell.2017.09.001. These authors deleted a distal enhancer of the developmentally regulated *Bcl11b* gene on mouse chromosome 12. This enhancer deletion results in localised changes in chromosome conformation (Referee 3 Fig. 3). We applied the Twins network trained on our own thymocyte Hi-C data to the Hi-C data of Isoda et al., 2017. Twins scores across chromosome 12 show a prominent peak around the location of the *Bcl11b* enhancer (108.4Mb), and provide a good reflection of visual changes to the Hi-C map (Referee 3 Fig. 3). This result shows that Twins can indeed identify small-scale differences that result from focal perturbation of the genome in developing T cells. This analysis is now included as Figure 2G in the revised manuscript.

Referee 3 Figure 3 (New Fig. 2G of the revised manuscript). Twins scores reflect focal perturbations in chromosome architecture.

Isoda et al., 2017 deleted a distal enhancer of the developmentally regulated *Bcl11b* gene on chromosome 12. The location of the enhancer at 108.4Mb and of the *Bcl11b* gene are shown. Enhancer deletion results in localised changes in the Hi-C map. Twins scores across chromosome 12 show a prominent peak at the location of the *Bcl11b* enhancer

3. The biological meanings of the features extracted by Twins were not very well illustrated. Although the authors showed some enrichment of CTCF, cohesin, and chromatin modifications in the different regions identified with Twins, how these differences correlate to

cellular function and previously identified conformation features like TADs and loops had not been carefully examined. For instance, whether the degree of embedding distances has specific indications for chromosome conformation? Could the author add some analysis to check how the embedding distances related to TAD insulation, compartment, and CTCF/cohesin binding in Figure 2B-2D? What are the genes being affected in these regions and whether they are critical for T-cell differentiation? Most important, in Figures 7 and 8, what are the biological meanings of the differential features? Did the authors find some differential features in T-cell differentiation system?

We thank the referee for this important point. In addition to the extensive analysis on the relationship between Twins scores and chromatin changes in T cell development discussed in response to this referee's point 2 above, Supplementary Figure 2 of the revised manuscript presents an illustration of chromatin changes in control and CTCF-depleted neuronal progenitor cells. Increased Twins distances correspond to visibly different Hi-C maps, the occurrence of deregulated genes, the density of CTCF motifs, and the density of CTCF ChIP-seq peaks (Referee 3 Fig. 4).

Referee 3 Figure 4 (New Supplementary Fig. 2 of the revised manuscript). Illustration that regions with high Twins scores after CTCF depletion in neuronal progenitor cells are characterised by changes in contact domains, deregulated genes, and a high density of CTCF motifs in addition to a high density of CTCF binding detected by ChIP-seq.

We did find differential features in the T-cell differentiation system. One example is provided by chromatin jets, which we recently described as a new feature in chromatin contact maps (Guo et al., 2022, DOI: 10.1016/j.molcel.2022.09.003). Twins found two instances of jets that were lost or gained during DP to SP differentiation (Reviewer 3 Fig. 5).

Reviewer 3 Figure 5: Twins finds chromatin jets regulated during T-cell differentiation.
 A. Twins embedding distances of the regions chr19: 29.6-32.16Mb and chr9: 116-118.56Mb are shown against the distribution for all regions along the Y axis, demonstrating that the condition embedding distances for these two regions are high.
 B. Feature extraction performed on the two regions depicted in A. demonstrating the presence of differential jets.

4. Another concern about Twins is the use of parameters for the image. Authors should provide certain rationales for the use of the parameters such as why the stride was used as 160 kb in a window size of 256xR? Whether the parameters will affect the performance? How robust the performance is? If possible, the authors should provide a guideline for choosing optimal parameters.

The window size is fixed as 256xR because 256x256 images are used routinely in computer vision, and proven network architectures are available. A good network architecture is important for the ability of the network to train and to find differential features.

The stride is the distance between each image observation along the diagonal of the interaction map. It is important for the stride to be less than the image size to ensure coverage of the full genome and a variety of features present in the train set at different positions relative to the centre of the image. However, increasing the stride density impacts the number of datapoints used at training which has three key consequences. The first consequence is that there is a higher potential for the network to overfit on the train data. The second is that train time will be longer, see table of train times and stride choices. Finally, as the stride becomes smaller the file size becomes larger. With multiple networks to train in a size limited environment this constraint can be a significant for most users, see table of file size and stride choices.

We have included this information now summarised in a paragraph in our materials and methods and as documentation to supplement our code base.

Stride density per window	stride size (kb)	Train time for each of 5 random seeds						File size (per file)
2	1280	2m12.378s	2m24.900s	1m32.333s	1m35.878s	1m35.831s	397MB	
4	640	3m53.511s	3m53.004s	3m8.922s	3m6.894s	3m7.922s	796MB	
8	320	7m39.682s	7m41.851s	6m10.678s	6m9.904s	6m16.464s	1.55GB	
16	160	15m53.314s	13m1.052s	12m49.414s	12m57.209s	12m1.787s	3.10GB	
32	80	31m51.314s	32m35.059s	27m13.123s	26m33.553s	27m4.180s	6.22GB	
64	40			~ 1hr			12.44GB	
128	20			~ 2hrs			24.88GB	

5. In Figure 6A-D, the authors claimed that Twins could learn sequencing depth-related characteristics of Hi-C data if differences in sequencing depth occur across conditions, which was quite confusing since we would expect no conformation differences if two datasets were only different in sequencing depth. Could the authors illustrate what are the sequencing depth-related characteristics? How do these characteristics differentiate from sequencing depth-independent characteristics? Overall, Figure 6 was not clear. Further explanations are needed.

Sequencing depth affects resolution and the number of visible features. For example, sub-TADs or contact domains only appeared as sequencing was performed more deeply. Similarly, the density of fine-scaled features such as loops can be impacted by the sequencing depth (Akgol Oksuz et al., 2021 DOI: 10.1038/s41592-021-01248-7). Low-resolution interaction maps have non-uniform noise scattered across the interaction map. This type of noise is much easier for machine learning algorithms to learn than complex features. Since some of the features we are interested in can also result from deeper sequencing, it is not possible to disentangle this noise. Manuscript Fig. 6 indicates that deliberate training on data with highly mismatched sequencing depths can result in networks that have learnt sequencing depth artefacts.

It is possible to visualise the manner in which a network places emphasis on different components of the interaction maps using the integrated gradients. Supplementary Fig. 6 illustrates this for a network that has learnt sequencing depth artefacts compared to a well trained network. We have added the following sentence to the revised manuscript to clarify this point: **“This is because discrepancies in sequencing depth can change both the sparsity and the fine-scaled features observed in chromatin contact maps”**.

6. All the validations were performed on chromosome 2 across the manuscript. The authors should provide at least one additional validation to show another chromosome. This is important if the conformation changes are chromosome-specific.

We thank the reviewer for this point. To address this concern we have trained an additional five T-cell differentiation networks (trained without chromosomes 1,3,4,5,6 respectively) and compared the results from each. We find that the choice of test data has little impact on the results, which are visually very similar regardless of the choice of test chromosome, and highly correlated genome-wide. These data are presented in Supplementary Fig. 1 of the revised manuscript.

Of note, users may want to split the genome by location as opposed to by chromosome, and our code base supports this choice.

New Supplementary Figure 1: The choice of test chromosome does not impact the Twins embedding distance.

A. Twins embedding distances for conditions (orange) and replicates (blue) are plotted for thymocytes at CD4 SP versus DP stages of differentiation are depicted across chr 1 for networks trained with a different choice of test chromosome.

B. The correlation coefficient for the genome wide embedding distances calculated using networks each with a different choice of test chromosome from chromosomes 1-6.

Minor comments:

1. In Figure 2B-2D, the distances around telomeric regions are very striking. Is it a common feature or a specific feature either related to specific conditions or specific chromosomes? Telomeres are hard to sequence and the resulting Hi-C maps are sparse. This can lead to high replicate distances in telomeric regions. The specific locations depend on the reference genome (mm9 for liver and thymocytes, mm10 for NPCs).

2. Figure 2A: the embedding distance plots didn't have a scale.

We thank the reviewer for noticing this oversight, we have now amended this in the manuscript.

3. Figure 4B: how the contact domains were defined? If the differential CDs were identified by another method, how did the distances identified by Twins coordinate with the differential CDs? It's would be better to include embedding distance plots for the selected regions.

We have revised the Figure legend to indicate that the embedding distances are included in Fig. 4C. Differential domains were quantified by a different method and this is now clarified in the legend as well as in in the materials and methods.

4. Figures S3 and 6: please consider using "high-depth" and "low-depth" other than "high-res" and "low-res", which were not appropriate since the resolution was the same.

We thank the referee for this recommendation, and we agree that the use of low-depth and high-depth are more appropriate. We have amended the manuscript accordingly.

5. Figures 7 and 8: What's the meaning of the red and blue regions in "isolated features" in Figure 7? What are the algorithms to obtain gained and lost features? How the "threshold" was defined?

These are defined using the Twins trained network. The threshold is placed based on the 0.95 quantile of same operation applied to the replicates. Red is gain between the conditions and blue is lost between the conditions this is now clarified in the figure legend. The following sentence is added to clarify the choice of threshold in the materials and methods "We calculate the threshold by taking the 0.95th percentile of values obtained by the application of the same method on the replicate map".

6. For Hi-C contact images in the manuscript (Figure S2C, S2D, S3B, S3C, S4B), please plot a density bar to show the contact numbers in the images.

We thank the reviewer for their attention to detail and have now included a density bar for the chromatin conformation maps requested.

Reviewer #1 (Remarks to the Author):

The authors have addressed all my concerns. As a foundation work, the downstream impact may be at different areas beyond our expectation, I would like the author to included all results done in this study (e.g. the loss function study as described in the response) in public access place (e.g. github or supplemental materials.) Therefore, other researcher can read the results.

Reviewer #2 (Remarks to the Author):

The authors have addressed all my comments.

Reviewer #3 (Remarks to the Author):

The authors have addressed my concerns and improved the manuscript according to the suggestions. The manuscript is now suitable to be published.